# *RLHF-Blender*: A Configurable Interactive Interface
# for Learning from Diverse Human Feedback

**Yannick Metz** [1]   **David Lindner** [2]   **Raphaël Baur** [2]   **Daniel Keim** [1]   **Mennatallah El-Assady** [2]

## Abstract

To use reinforcement learning from human feedback (RLHF) in practical applications, it is crucial to learn reward models from diverse sources of human feedback and to consider human factors involved in providing feedback of different types. However, the systematic study of learning from diverse types of feedback is held back by limited standardized tooling available to researchers. To bridge this gap, we propose *RLHF-Blender*, a configurable, interactive interface for learning from human feedback. *RLHF-Blender* provides a modular experimentation framework and implementation that enables researchers to systematically investigate the properties and qualities of human feedback for reward learning. The system facilitates the exploration of various feedback types, including demonstrations, rankings, comparisons, and natural language instructions, as well as studies considering the impact of human factors on their effectiveness. We discuss a set of concrete research opportunities enabled by *RLHF-Blender*. More information is available at our Website.

## 1 Introduction

Reinforcement learning from human feedback (RLHF) is a powerful tool to train agents when it is difficult to specify a reward function or when human knowledge can improve training efficiency. Recently, using multiple forms of human feedback for reward modeling has come into focus (Jeon et al., 2020; Ghosal et al., 2023; Ibarz et al., 2018; Bıyık et al., 2022a; Mehta & Losey, 2022). Using diverse sources of information opens up several possibilities: (1) feedback from different sources allows for correcting poten-

tial biases in the data; (2) the feedback type can be adapted to a particular task or user based on preferences, knowledge state, or available input modalities; (3) agents can actively select an appropriate type of feedback during training to optimize learning. However, implementing such systems is challenging because it requires an interplay of different disciplines, such as machine learning, user interface design, and even psychology. A particular limitation is that there are no readily available tools to collect and learn from diverse types of human feedback. Consequently, there has been little systematic investigation of learning from a larger set of feedback types.

Moreover, existing estimates of human feedback quality used in reward modeling are often based on simplified assumptions (Ghosal et al., 2023). Grounding assumptions on the quality of human feedback in data from human subject studies can improve the learning of human-aligned reward models.

To enable such studies, we propose *RLHF-Blender*, which provides a standardized and modular setup that enables systematic empirical studies with human subjects to study learning from different feedback types (Figure 1). *RLHF-Blender* generalizes across various possible environments and users and enables a detailed analysis of various dependencies to improve estimates of human feedback characteristics like irrationality or bias. Our prototype implementation showcases a series of user interactions for different types of human feedback and how they can be processed as input for reward models. We aim to align the efforts of machine learning and human-computer interaction towards a goal of expressive and versatile human feedback for human-AI interaction.

**Contributions** — In this paper, we (1) Highlight challenges and possible solutions for learning from diverse human feedback. We discuss how considering different dependencies can improve the estimation of human feedback quality. When then present a standard encoding scheme for human feedback ; (2) Introduce *RLHF-Blender* as the implementation of an interactive application for the investigation of human feedback in reinforcement learning; (3) Discuss potential future research opportunities to learn from diverse human feedback enabled by our system. *RLHF-Blender* will be made available for research as open-source software.

---

[1]Department of Computer and Information Science, University of Konstanz, Germany [2]Department of Computer Science, ETH Zurich, Switzerland. Correspondence to: Yannick Metz <yannick.metz@uni-konstanz.de>.

*Interactive Learning with Implicit Human Feedback Workshop at ICML 2023*, Honolulu, Hawaii, USA, 2023. Copyright 2023 by the author(s).

*Figure 1.* An overview of *RLHF-Blender*: The user interface serves samples of agent behavior to the user. These samples are drawn from an episode buffer with steps from an online RL agent or offline data sources. Via the user interface, users can provide different types of feedback. *RLHF-Blender* translates these different types of feedback into a standard encoding, which is then passed to a reward model. All user interactions and feedback can be logged and stored for post-hoc data analysis and reward model training.

## 2  Related Work

In this section, we provide a brief overview of work on reinforcement learning from human feedback.

**Reinforcement Learning from Human Feedback —** Using human feedback as the sole or an additional source of reward information has gained traction in research (Ng et al., 2000; Knox & Stone, 2009; Griffith et al., 2013; Christiano et al., 2017), especially for the alignment of large language models (Ouyang et al., 2022). This paper focuses on learning rewards solely from human feedback, excluding approaches like interactive reward shaping (Knox & Stone, 2009; Warnell et al., 2017). Different types of feedback, ranging from ratings, demonstrations (Ng et al., 2000; Abbeel & Ng, 2004), comparisons (Wirth et al., 2017) to interruptions (Hadfield-Menell et al., 2017) or even language and narrations (Fish et al., 2018; Sumers et al., 2022b) has been proposed. A wide range of different types of feedback can be interpreted via a common framework of reward-rational choice (Jeon et al., 2020).

**Modeling Irrationality and Bias —** Initial research in learning from human feedback worked with the assumption of optimal human feedback (Ng et al., 2000; Abbeel & Ng, 2004), but it has been established that human feedback is generally sub-optimal and noisy (Ramachandran & Amir, 2007; Ghosal et al., 2023). Models like a Boltzmann distribution capture human irrationality in decision-making (Luce, 2012) and feedback (Ziebart et al., 2010; Jeon et al., 2020). However, such models are often based on simplified assumptions. Studying the characteristics of feedback in human studies can improve the estimation of irrationality and bias and, in turn, learning based on these reward models.

**Human-Centered Studies —** Despite calls to ground models of human feedback in human-subject studies instead of simplified heuristics (Ghosal et al., 2023; Shah et al., 2019), so far, the investigation of human factors in learning from feedback, in particular from different types, has been limited. One existing study explored the interaction of 20 participants with a robot via different input channels (Koert et al., 2020) in a simulated kitchen and a sorting task. They

uncovered a positive effect on agent performance and challenges like the intuitiveness of different feedback types or frustration if the robot ignores human feedback. In a different study (Bıyık et al., 2020) with 27 total participants, the correct formulation of queries was investigated. This study showed that an information-gain objective decreasing uncertainty was well suited to propose effective questions. There has been a recent effort to provide testing environments for human-subject experimentation in reinforcement learning (Taylor et al., 2021) or active querying (Bıyık et al., 2022b). However, existing work has focused on simple feedback types and user interfaces, and more extensive human studies as possible future research directions have been highlighted.

## 3  Learning from Human Feedback

**Preliminaries —** In this work, we are interested in analyzing human feedback used for reward learning for training machine learning agents. Our discussion is not restricted to a specific implementation of reinforcement learning from human feedback (Christiano et al., 2017; Wirth et al., 2017). We assume that an agent can learn from a set of observations generated in an online learning process from the agent acting in an environment or based on offline data from recorded behavior of, e.g., humans or a trained ML model. In general, the reward function for a task is fully specified by human annotators. In certain conditions, a ground-truth reward function can be available for the complete or part of the observation space. This allows us to compare human feedback with such ground-truth rewards. Additionally, we will discuss the possibility of calibrating estimates of human feedback quality based on a ground-truth reward function.

### 3.1  Dependencies Influencing Feedback Quality

Previous work has pointed at differences in human irrationality for different types of feedback (Ghosal et al., 2023). There are a series of various factors and dependencies that can influence the quality of human feedback.

We start with dependencies that can be directly disentangled and analyzed via human subject studies:

**Type-Dependency** — As has been noted before, quality might differ between types of feedback, e.g., comparisons might have a higher quality compared to ratings, etc. (Ghosal et al., 2023).

**Task-Dependency** — We can generally assume that the feedback quality is dependent on the task, e.g., a tasks observation space-dimensionality, complexity, diversity of possible (close-to)-optimal policies, among other characteristics.

**Progress-Dependency** — Feedback can differ in quality at different training phases, e.g., at the beginning or with a converged model policy (Arzate Cruz & Igarashi, 2020; Knox et al., 2012).

Additionally, there are additional dependencies that may influence feedback but are only partially measurable because they cannot be fully disentangled from other dependencies:

**Personality-Dependency** — Feedback can depend on a human's personality or background, which might influence the quality under different criteria. This might encompass things such as directness or frustration tolerance (Arzate Cruz & Igarashi, 2020; Lindner & El-Assady, 2022).

**Knowledge-Dependency** — A human's ability to give high-quality feedback can depend on their knowledge, i.e., if the task is of high complexity or requires domain expertise.

**Demand-Dependency** — Cognitive demand and effort can play a significant role in the quality of feedback, particularly if we might expect feedback to be incomplete or inaccurate due to high effort (Knox et al., 2012).

These, and potentially additional dependencies, can be the target of experimental evaluation, and understanding them better can improve reward modeling in the future. Therefore, we present a dynamic and flexible experimentation environment that allows us to cover a large space of tasks, types of feedback, and users.

## 4 Human Feedback Types

We strive to enable learning from diverse human feedback. To enable this, we first outline a structuring scheme that captures a wide range of possible human feedback. This structuring leads to a standard encoding format for arbitrary feedback. We then discuss how five commonly used types of human feedback can be modeled can be interpreted via the encoding. In the following chapter, we present interaction mechanics for these feedback types and describe how our software framework interprets them.

### 4.1 Structuring Human Feedback

We structure incoming human feedback according to six core dimensions: (1) the *granularity of feedback*; (2) if it targets *observed* behavior or includes *generated* ; (3) if it

targets a *single* observation/generation or contains a *relative statement*; (4) if it is given for *instances*, i.e., particular states and actions, or general *features*; (5) what actual feedback information is available based on the *human intent*; and, (6) how the feedback was *expressed*. In the following, we describe the attributes required in common feedback encoding.

Observations can be provided from different sources, such as recorded state-actions sequences from online agents acting in the environment, offline data, demonstrations generated by a human expert on the fly, etc. We can give feedback at different levels of **granularity**, e.g., for the entire episode level or a single state, etc. Each encoding instance of feedback thus contains a reference to a single or a set of episodes/states/segments, which we call the *targets $T$* of a feedback instance. The target reference allows us to assign feedback to available state-action pairs. In one special case, feedback might not be targeted at any particular set of state-action pairs, which might be indicated by a unique value and can be handled by a downstream reward model.

Many types of human feedback can be assigned to targets **observed** by the human, i.e., a rating or preference comparison of samples shown to the human. However, feedback can also be **generative**, i.e., containing new behavior generated by humans as demonstrations or desired goal states. From a modeling perspective, we can encode generated observations equivalently to existing ones by integrating them as target references. However, as generated behavior is not sampled from the same underlying policy as observed behavior, it should be treated accordingly.

Thirdly, feedback might be defined for a single, **absolute** target, for example, a rating or desired goal state or a **relative** statement, e.g., a ranking or comparison. Such a relative statement is expressed via a partial ordering $T_1 \leq T_2 \leq T_3, ...$, encoded via a list of targets and respective ranking indices, e.g., 1 and 2, in a pairwise comparison. If no partial ordering is defined, e.g., for a batch of demonstrations, we assume each target to be absolute and split batches of feedback into absolute elements.

Feedback might be given for **instances**, i.e., entire observations, or on a **feature**-level, i.e., for abstract features. Feature-level feedback might highlight important or desired features via techniques like gaze tracking or asking humans for desired goal states. When constraining feature-level feedback to a subset of the observation space, we can save this feedback in the same format. However, we add a flag to indicate feature-level instead of instance-level feedback.

After establishing the *feedback target*, we now need to classify the information humans give based on their **intent**. We use the categories of *evaluative*, *instructive*, *descriptive*, and non-intentional feedback. For evaluative feedback, we assume a score is available according to the target granular-

ity, e.g., a rating for an episode or a comparative ranking between features. Instructive feedback, like demonstrations or corrections, contains a reference to a target, potentially a confidence/optimality score, and often either a full action distribution or single action assigned to a state, assumed to represent the human-optimal policy. Lastly, descriptive feedback contains feature-level information or annotations targeted at the entire tasks and environment.

Finally, feedback might either be **expressed** explicitly, i.e., via conscious direct human input, or implicitly, i.e., indirect, potentially subconscious expressions like gazes or mimics. While it is not feasible to implement a universal encoding that directly fits raw inputs from each available implicit feedback type, our encoding supports the passing of extracted scores, features, or instances alongside a potential quality/confidence score that encodes the estimated extraction quality from an extraction model. However, additional features and meta-data might be passed alongside each feedback as input for custom downstream reward models.

**A Standard Encoding for Human Feedback —** We utilize these dimensions to classify feedback into standard encoding. We combine each feedback instance with additional metadata like timestamps, verbal meta-level comments, uncertainty, etc., enabling detailed downstream analysis. We summarize the standard encoding as an abstract grammar:

$$FB \rightarrow \; < T|\{T_1, T_2, ...\}, CL, Info, \textit{Meta-Data} >$$
$$T \rightarrow Episode|State|Segment|All$$
$$Info \rightarrow Evaluate|Instruct|Describe|None$$
$$Evaluate \rightarrow score$$
$$Instruct \rightarrow Policy|Action$$
$$Describe \rightarrow Mask|State, Annotation$$
$$None \rightarrow ...$$
$$CL \rightarrow Instance|Feature$$
$$\textit{Meta-Data} \rightarrow Timestamp, User\text{-}ID, ..$$

### 4.2 Feedback Types

To outline the utility of our proposed system and framework, we choose five exemplary types of feedback and outline user interactions, how these types of feedback can be encoded, and finally, comment on how a reward model might interpret them. In subsection 5.2, we discuss an implementation for a user interface enabling these types of feedback.

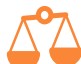 **F1: Evaluative Feedback —** We define this type of feedback for cases in which a human gives a numerical or otherwise quantifiable judgment of a target, e.g., binary feedback or a numeric score (Arzate Cruz & Igarashi, 2020). It is generally defined for a single target (simultaneous feedback for a group of targets can be split into single-target feedback by assuming conditional inde-

pendence). It may be given at different levels of granularity; common are evaluative feedback for an entire episode or for single steps (Knox & Stone, 2009; Griffith et al., 2013). As users, we must be able to select a clearly defined target and rate it via some numerical or ordinal input. This type of feedback is comparatively easy to utilize for reward models because it can be used as a prediction target.

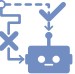 **F2: Comparative Feedback —** Here, a user makes a relative judgment, i.e., a pairwise comparison between two targets or a ranking of multiple targets. This type of preference-based feedback is widely used because it is often easier for humans to give comparative judgment compared to absolute scores (Wirth et al., 2017). Comparative feedback is a set of targets with an associated ordering relation (e.g., rank values). A common granularity is segments of states-action sequences (Wirth et al., 2017; Christiano et al., 2017). For comparative feedback, users must be able to select a set of targets and express order or preference via their input. Rankings of targets need to be translated into scores as prediction targets for reward models.

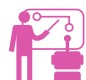 **F3: Corrective Feedback —** Here, the user has a trajectory showcasing imperfect behavior as a reference and needs to instruct via an improved strategy. This can be done either implicitly, e.g., by pushing a robot into a correct position (Mehta & Losey, 2022), or explicitly via specifying a better action. Implicit corrections require an additional step translating the corrected trajectory into a sequence of agent actions. Common granularities are corrections for single states or short segments. Corrections are given as either a preferable policy distribution or optimal action for each target. User interactions implicit interactions enable correct behavior, e.g., by dragging or allowing the user to specify a correct action in a state. Corrective feedback can be used to directly optimize the behavioral policy of an RL agent or translated into comparative feedback, with the corrective trajectory as the preferred trajectory.

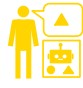 **F4: Demonstrative Feedback —** Here, the human is asked to provide a reference of optimal behavior that the agent should imitate (Ng et al., 2000). Like corrective feedback, it is instructive, providing actions for a sequence of generated states. This type of feedback generally requires an environment-specific user interface to generate demonstrations. In some cases, e.g., in continuous control environments, it may not be feasible for humans to generate demonstrations. Similar to explicit corrective feedback, we can use demonstrations directly to optimize the policy via supervised learning or assign demonstrated behavior the highest possible reward to optimize the reward model.

**F5: Descriptive feedback —** While there are different ways of defining descriptive feedback, we chose a formulation in which descriptive feedback is a qualitative judgment about features (Sumers et al., 2022a). For example, certain feature values might be desir-

able to achieve a certain (sub-)goal. Therefore, specifying a (partial) goal state could be treated as descriptive feedback. To enable this type of feedback, a system must provide a way for users to annotate or generate inputs in the feature space, e.g., by highlighting important features or generating desired goal states). Descriptive feedback could be treated as a constraint during reward model optimization (e.g., the presence of features should lead to a higher predicted reward).

# 5 An Implementation for Reinforcement Learning from Different Feedback Types

This section describes our system implementation, starting with an overview and then discussing sub-components.

## 5.1 System Overview

*RLHF-Blender* (see Figure 1) consists of three major components: (1) An interactive user interface that enables browsing episodes or segments from a set of available state-action sequences and implements multiple feedback interactions which can be enabled or disabled dynamically, (2) a feedback processor consisting of a sampling and a translator unit, to serve appropriate episodes/segments to the user, and to translate human feedback to a standardized format, (3) a consistent software interface to train reward models with the episode data and human feedback. All three components are highly modular to enable different combinations of modules for versatile human-subject studies.

The system enables different training configurations: It can be used with **online** data, i.e., an RL agent is trained synchronously with a reward model. During an experiment session, a reward model and agent can be trained on the human feedback reward model, which in turn allows the sampling of new trajectories by rolling out the online policy. However, such a setup might require careful synchronization or tuning. Therefore, the preferred configuration is an **offline** mode, which uses pre-collected episode data to train reward models. A suitable dataset, e.g., contains trajectories generated by agents of different skill levels. One can dynamically serve episodes of different skill levels during an experiment session. Alternatively, episodes labeled with a ground-truth reward function allow the sampling of increasingly higher-skill behavior, potentially adapted to human feedback performance.

## 5.2 User Interface

Our application provides a comprehensive user interface, which provides a series of visual interfaces and interactions for different feedback types. The interface can be dynamically configured in the experiment design stage, e.g., enabling or disabling different types of feedback and UI elements. The interface with the feedback interactions is shown

in Figure 2. The interface enables five separate interactions which can be dynamically enabled or disabled: Ratings, so evaluative feedback targeted at observed episodes, can be given via a slider, which can be adjusted to allow for different granulates, i.e., only binary or more fine-grained ratings on a scale. Comparisons or rankings are implemented via a drag & drop interaction that allows users to put a set of episodes in a desired order. Again, the number of possible elements can be dynamically adjusted, e.g., only to enable pairwise comparisons. Furthermore, for vision-based applications, shown in Figure 2 is the *BabyAI* (Chevalier-Boisvert et al., 2019) RL environment, brushing allows users to specify feature importance. By brushing a region in the image, important or unimportant regions can be highlighted, which can be passed onto an agent as additional information (Guan et al., 2021). Finally, it allows the integration of additional modules to generate demonstrations and action advice. Humans can select a section or single state of episodes to give corrective feedback, e.g., by inputting a better action compared to the one taken by the agent. For demonstrations or corrections, the interface allows to integrate environment-specific interfaces.

This primary interface can be enriched with additional interaction mechanics and visualizations, e.g., to investigate possible explainability methods. Our system allows the user to scroll through the entire dataset of existing episodes, e.g., all the historically generated trajectories by current and past agents. Figure 2 shows a scroll bar on the left side that enables episode selection, as well as highlighting of already labeled episodes or potentially high-impact episodes (indicated via color, in this case, purple).

The bottom-right corner of Figure 2 showcases the possibility of integrating views for additional information. In this case, it presents a user with an estimate of their feedback's quality or success, e.g., by comparing a ground-truth model without human reward with the performance of a user-trained reward model. We plan for the system to have the ability to integrate views that can showcase various types of information, or even explainability tools, that can support the human and tackle issues such as lack of feedback and resulting frustration.

## 5.3 Feedback Processor

A second important component is the *feedback processor*, which handles both the processing of incoming human feedback of different types and the selection of episodes served to the human, similar to querying the human for feedback at a particular point in training.

### 5.3.1 TRANSLATOR

A *translator* component receives potentially heterogeneously structured feedback from the user interface. The

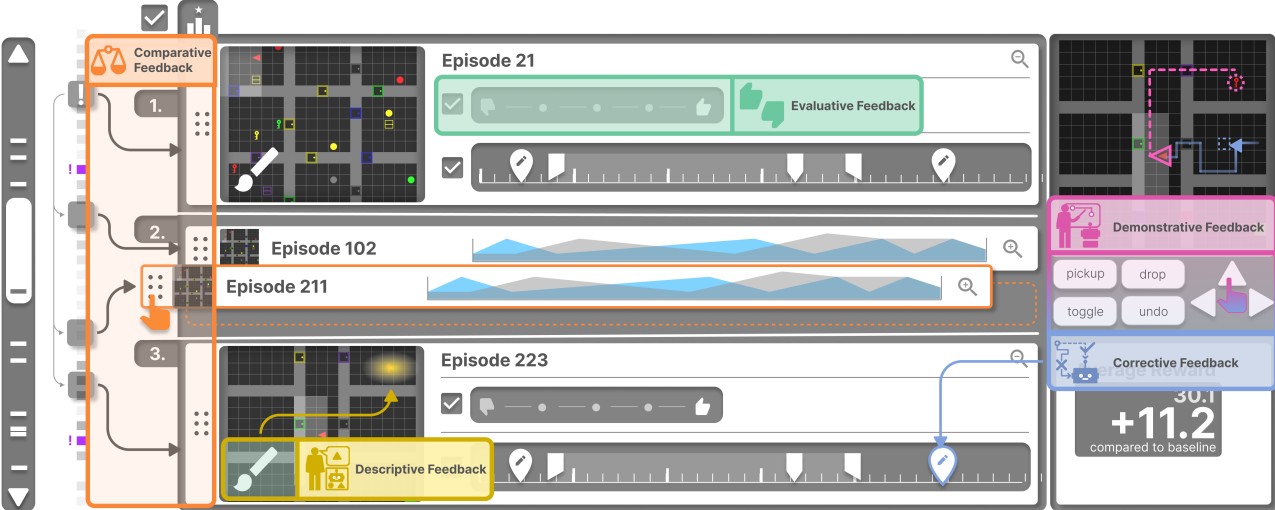

*Figure 2.* A possible realization of the interface enabling all described feedback types. 🔀 Users can rate individual episodes using the interactive Likert scale. ⚖ Users provide comparative feedback by dragging and dropping episodes into different categories. 🔧 For a selected episode, the interactive control element allows users to correct actions in previously sampled episodes. Here, the agent makes an unnecessary turn, and the user corrects it by selecting the corresponding point in the trajectory and suggesting another action. Throughout the interface, visual hints encourage the user to correct specific actions. 🎮 The same control element is used to generate demonstrations from scratch, e.g., how to reach the next crucial item in this environment. 🧑 The brush tool allows users to highlight crucial parts of the agent observation to provide descriptive feedback.

translator depends on the standard feedback encoding presented in subsection 4.1 as the data storage format for translated feedback. This standardized feedback can be used to train single- or multi-type reward models. The translator component combines and matches feedback sent from the user interface with the respective segments, meta-data, etc.

### 5.3.2 SAMPLING/QUERYING COMPONENT

A *sampling* component determines which samples to show in the user interface, i.e., for which samples we query the user for information. Depending on the desired setup, we might use different sampling modes, ranging from passive to active. We can configure the user interface to enable manual episode selection by the human, i.e., by providing an overview of the data buffer with the option to analyze samples in detail. This does not involve programmatic sampling. Alternatively, we provide three sampling modes:

- Random: Randomly sample episodes from the replay buffer without regard for the quality or diversity of shown samples. This might serve as a valuable baseline to compare against more advanced sampling strategies.
- Progressive: If an ordering of the data buffer is available, e.g., by a ground-truth reward function or order of generation by an online RL agent, we can choose to progressively sample increasingly high-quality or new samples to simulate training progress.
- Query-Based: We may define more advanced metrics

to select the most appropriate samples and perform targeted queries. For example, one can select samples with a high reward model loss if a reward model is trained with feedback in the background.

- State-Machine: We can switch sampling strategy throughout the process, e.g., starting with random sampling, then going over to query-based sampling, or changing data sources throughout the experiment. State-machine behavior can be combined with the other modes based on set criteria.

The sampling component is designed modularly to integrate potential advanced query mechanisms and sampling strategies for experimentation. This allows us to analyze the effect of different strategies on human behavior, reward model training, or online training of RL agents.

### 5.4 Reward Model

Based on the trajectory data and collected human feedback, we train reward models. Our system is directly integrated with the library of reward networks of the *imitation* package (Gleave et al., 2022), which simplifies compatibility with different algorithms and avoids code duplication.

We need to specify input data and a loss function for reward model training. Input data is available in the standard feedback encoding presented above, which states/actions etc., served from the buffer as model input. A developer can choose a loss function dynamically based on the available

feedback data. For example, suppose numeric evaluative feedback is available. Then, we can train a reward model with a simple regression loss like *mean squared error* to estimate a human-optimal reward function for state and potential actions. If both a ground-truth reward function and feedback with a single state granularity are available, we might also choose a shaping reward model to predict a policy-shaping term (Knox & Stone, 2009; Warnell et al., 2017). For different types of feedback, particularly instructive or comparative feedback, we might choose a different loss function to optimize the reward model (Christiano et al., 2017).

Again, the library is designed modularly, allowing the integration of different reward models or loss functions. Furthermore, suppose multiple feedback types are used simultaneously. In that case, one may either apply different losses to the same underlying reward model or aggregate the prediction from multiple reward models, e.g., via weighting or voting. Furthermore, additional data passed via the standard encoding, like confidence scores or meta-data, are available at training time and might be used as additional inputs, constraints, or ways to adapt the loss calculation.

### 5.5 Post-hoc Analysis

Finally, a key objective is facilitating post-hoc analysis, i.e., analyzing the quality and dependencies of human feedback in reinforcement learning scenarios. Therefore, our system puts emphasis on detailed logging and persistent storage of results. The standardized format allows the sharing and reusability of analysis scripts and tools.

Experiments are logged into separate files or tables, which enables the analysis of individual, anonymous users. Alongside the logged metadata and quantitative evaluation based on reward model training or downstream training of RL agents based on the learned reward model, we can investigate factors outlined in section subsection 3.1.

## 6 Research Opportunities for Diverse Human Feedback

We want to outline four possible study designs that can be realized with our system.

### 6.1 Reinforcement Learning from Human Preferences

Our system suits itself to replicate the well-known setup of reinforcement learning from human preferences (Christiano et al., 2017). During setup, we can configure a minimal user interface, potentially just with a progress bar and the ranking interface (see Figure 2D). Other types of feedback are, therefore, disabled. We restrict the number of displayed episodes to two for strict pairwise comparison. Furthermore, during the setup of the interactive user interface,

we can select additional instructions or info displayed to the participant when starting the experiment.

RL from human preferences is often implemented asynchronously, i.e., with a (pre-trained) RL agent being optimized based on an existing version of the reward model. Trajectories generated in this process are then saved in the data buffer. Then human preferences are collected based on the collected data. Here, we might choose either a random or progressive sampling of the buffer elements. A reward model is then optimized via supervised learning on the dataset of rated comparisons. Here, we may choose a simple neural network-based reward model which receives single states as an input and outputs a single scalar reward estimate. We then choose a loss function that optimizes a score function from pairwise preferences (Christiano et al., 2017).

### 6.2 Investigating Effectiveness of Multi-type Feedback

To investigate the effectiveness of simultaneous multi-type feedback, we enable multiple feedback types simultaneously, for example, ratings, ranking, and correction/action advice. For each episode, the participant can then choose which feedback type to use, which could enable the analysis of preferential types of feedback for different users. Alternatively, one can instruct the user to utilize multiple types of feedback for the same samples, which allows for experimenting with inter-modal validation or calibration of reward estimation methods. Combining numerical ratings and rankings of episodes could increase the expressiveness of reward models compared to pure pairwise comparisons.

The system automatically handles the translations of different feedback into the standard encoding and subsequent logging. However, in the case of multi-type feedback, the choice of the reward modeling approach is more complex. We might train a separate reward model with an individual loss function for each feedback type and then treat the trained models as a type of ensemble that might be combined via custom voting or weighted averaging. Alternatively, we might optimize a single model via a multi-objective loss that incorporates the different types of feedback. Finally, we can utilize frameworks like reward-rational choice (Jeon et al., 2020) as a common loss function. Comparing these different approaches, therefore, is a main research opportunity.

### 6.3 More Accurate Estimations of Human Irrationality

Humans might not be able to fully express their internal (user-optimal) reward function via feedback, i.e., the feedback is subject to noise or uncertainty stemming from an underlying human irrationality (Ghosal et al., 2023). In models such as the reward-rational choice framework (Jeon et al., 2020), we interpret instances of human feedback as imperfect expressions of an underlying reward function. Given a choice from a set of possible choices $c^* \in \mathcal{C}$ and a ground-

ing function $\psi : \mathcal{C} \to f_{\Xi}$ with $f_{\Xi}$ being the set of distributions over all possible trajectories of an agent, we define a *Boltzmann-rational* policy as (Jeon et al., 2020):

$$\mathbb{P}(c * | r_U, \mathcal{C}) = \frac{\exp(\beta \cdot \mathbb{E}_{\xi \sim \psi(c*)}[r_U(\xi)])}{\sum_{c \in \mathcal{C}} \exp(\beta \cdot \mathbb{E}_{\xi \sim \psi(c)}[r_U(\xi)])}$$

Here, the coefficient $\beta$ indicates the rationality, i.e., the certainty or estimated quality of a choice, and in extension, the human feedback. As pointed out by Ghosal et al., most work chooses a value for this coefficient based on assumptions, especially for simulated human feedback, which uses this formulation. The authors, therefore, propose to fit this $\beta$-coefficient to data for different types of feedback, like comparisons, demonstrations, and interruptions. They use an initial calibration phase with a known calibration reward function, similar to the previously described ground-truth reward function, to estimate the coefficient's value for different types of feedback. However, the authors point to possible limitations of such a calibration approach.

*RLHF-Blender* can serve to investigate characteristics of different human feedback types across various user groups, tasks, and training phases. Empirical results could serve as guidelines or priors for future estimations of human reward using the reward-rational choice or similar frameworks. Referring to our motivating example of the $\beta$-parameter in the Boltzmann distribution, we may assume that this $\beta$ is influenced by a range of different dependencies at a particular task, feedback type, point during training, and user (see subsection 3.1). For simplicity, we propose to interpret the value as a linear combination of these different dependencies:

$$\beta = \sum \alpha_{type}\beta_{type} + \alpha_{task}\beta_{task} + \alpha_{progress}\beta_{progress} + ...$$

with $\alpha$ as potential weighting factors, and $\beta$ being estimated for a specific setup. As a simplified assumption, we can, for example, assume a uniform weighting, i.e., $\alpha = 1/K$, with k being the number of considered factors. An estimate of different dependencies would allow us to dynamically adapt the irrationality estimate during training, fitted to the specific task, type of feedback, current progress in the training, and additional factors. We can then calculate the value of the different dependencies by setting a particular option fixed, e.g., the feedback type, and then marginalizing over all other dependencies. However, this is generally infeasible, particularly when adding more and more dependencies. Therefore, we advocate for dynamic and flexible experimentation environments that cover a large space of tasks, types of feedback, users, etc.

### 6.4 Training with Continuous Calibration

Using just a single calibration at the beginning of the training, as we have just described (Ghosal et al., 2023), might be insufficient for a full calibration. As outlined in section

3.1, many possible influence factors exist. Therefore, we could extend these calibration experiments via multiple distinct calibration phases or a continuous calibration where we purposefully sample examples with a ground-truth reward function or query the human for repeated feedback on the same samples to investigate properties like consistency.

To enable this experimentation, the main necessary design choice here is an appropriate sampling component. For example, to enable multiple calibration phases, we might choose a state-machine sampling, which switches modes based on pre-defined time intervals, which changes the underlying buffer from unlabeled online data to an offline buffer with a ground-truth reward available. An experimenter can also implement a custom sampling module based on the given system and data types, which intermixes samples of different data sources dynamically.

### 6.5 Investigating the Effect of Explainability

Finally, an interesting research question arises regarding the potential role of explainability methods and additional visual information for human feedback. Such additional information could improve the quality of human feedback or help the user to identify episodes that can profit from feedback in reward learning or reward shaping. Therefore, our system is designed to include additional visualizations or values as widgets in the user interface. This enables comparative studies to study the effect of explainability techniques on feedback quality, human confidence, and satisfaction.

## 7 Conclusion

As our system is still under development, we have not yet fully evaluated and deployed our system in a realistic environment, e.g., a large-scale user study. This also includes other researchers' applications of the system for custom experiments. While the system is designed to be highly modular, we might find that the current system design still requires substantial modifications and custom components or restricts the system's applicability. Therefore, validating the system design in contact with potential users is an important next step. We plan to make our system widely available to the research community as an open-source project.

In this paper, we have presented a framework and implementation for a configurable interface enabling learning from diverse human feedback. We have presented possible dependencies that influence human feedback and should be empirically investigated. We have then introduced *RLHF-Blender* as a modular system that enables experimenting with different configurations for user interfaces, feedback processing, and reward modeling, thus enabling highly flexible setups. *RLHF-Blender* is available at `https://rlhfblender.info`.

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

# A   Workflow

In the following section, we want to present the current workflow for potential users of *RLHF-Blender*. In particular, we outline how an experiment configuration can be created and show how the resulting data is logged and processed.

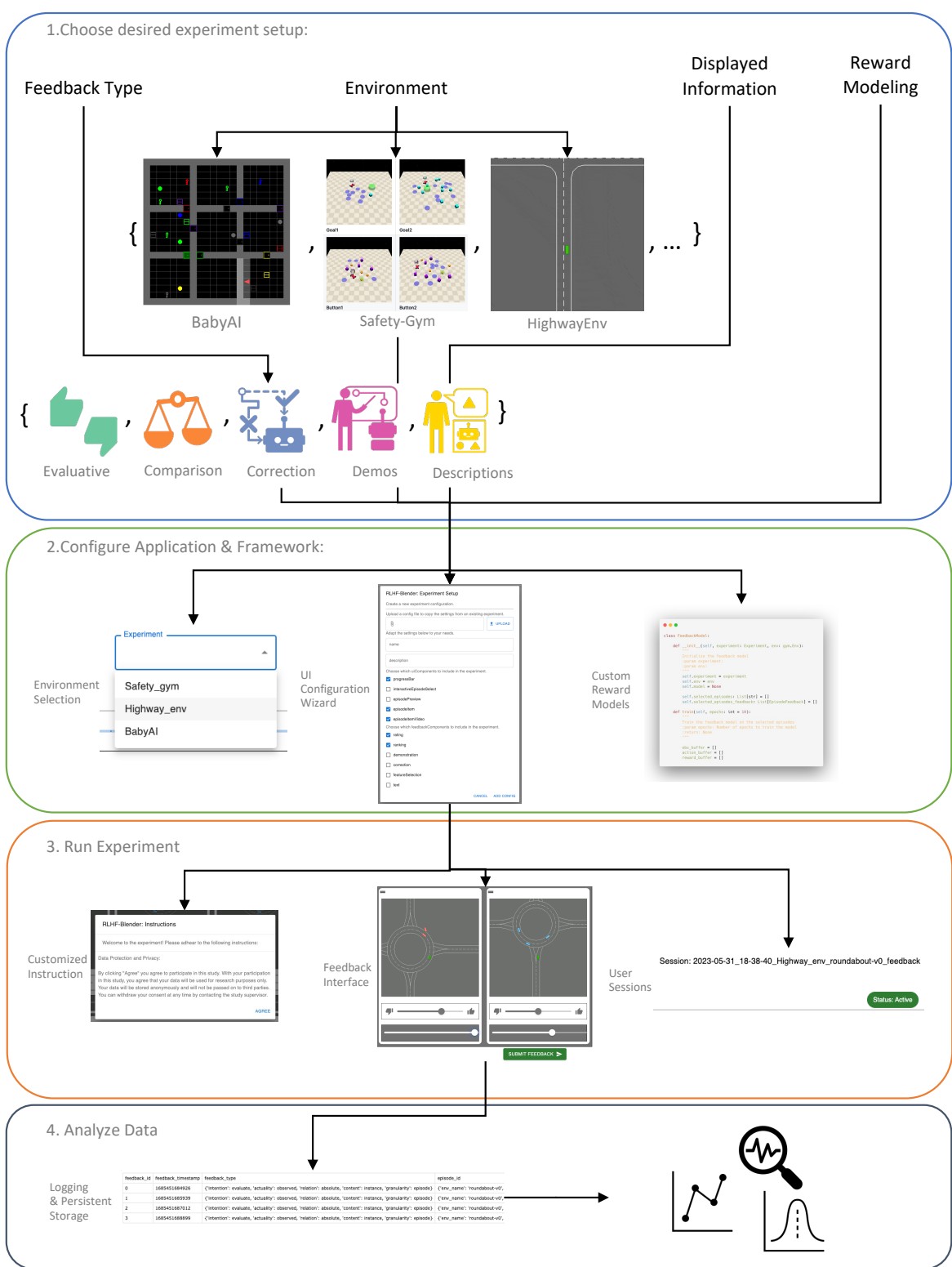

# B  Technical Specifications

*RLHF-Blender* is implement as a client-server application, with a FastAPI (Tiangolo, 2023) Python web-server and a React-based (Facebook, 2023) user interface.

The server is designed to integrate different sub-components, such as custom sampling, translation, or reward models. The framework utilizes data formats compatible with widespread libraries *Gym* and *StableBaselines3* (Raffin et al., 2021). This ensures that we can support various RL environments, algorithms, and data structures and that researchers can integrate our system into established workflows. Furthermore, the library is directly compatible with the library of reward models included in the Imitation library (Gleave et al., 2022).

The user interface is extendable via custom React components, e.g., to integrate custom explainability or information visualizations. In particular, an interface for custom demonstration-generation components is available. Such components could be created for specific RL environments. As a template, a demonstration generation interface for simple navigation tasks used for BabyAI (Chevalier-Boisvert et al., 2019) is provided.

*RLHF-Blender* can be deployed as a containerized application, potentially connected to a shared data storage for feedback, e.g., an SQL database. This setup allows the application to scale to many clients and servers and could therefore enable larger-scale distributed human experiments.

# C  Code Example for Standardized Data Format

The following code example showcases the described standard feedback encoding in Python code. Here, *StandardizedFeedback* serves as the container for arbitrary feedback that can be specified via the existing options.

```python
import enum
import gym
from pedantic import BaseModel
from typing import Union, List

class EpisodeID(BaseModel):
    """
    Reference to a specific episode saved in a data buffer
    """
    env_name: str = ""  # e.g.: BreakoutNoFrameskip-v4
    benchmark_type: str = ""  # e.g.: trained
    benchmark_id: int = -1  # e.g.: 1
    checkpoint_step: int = -1  # e.g.: 1000000
    episode_num: int = -1  #

class FeedbackDimension(enum.Enum):

    def __str__(self):
        # just return the enum value
        return self.name

    def __repr__(self):
        # just return the enum value
        return self.name

class Intention(FeedbackDimension):
    evaluate = 1
    instruct = 2
    describe = 3
    none = 4

class Expression(FeedbackDimension):
    explicit = 1
    implicit = 2

class Actuality(FeedbackDimension):
```

```python
        observed = 1
        hypothetical = 2

class Relation(FeedbackDimension):
    absolute = 1
    relative = 2

class Content(FeedbackDimension):
    instance = 1
    feature = 2

class Granularity(FeedbackDimension):
    state = 1
    segment = 2
    episode = 3
    entire = 4

class Target(BaseModel):
    id: int = -1
    origin: str = "replay"
    timestamp: int = -1

class Episode(Target):
    reference: EpisodeID = None

class State(Target):
    reference: EpisodeID = None
    step: int = -1

class Segment(Target):
    reference: EpisodeID = None
    start: int = -1
    end: int = -1

class StandardizedFeedbackType(BaseModel):
    intention: Intention = Intention.evaluate
    actuality: Actuality = Actuality.observed
    relation: Relation = Relation.relative
    content: Content = Content.instance
    granularity: Granularity = Granularity.episode

class Evaluation(BaseModel):
    rating: Union[float, int, List[float], List[int]] = None
    comparison: Union[float, int, List[float], List[int]] = None

class Instruction(BaseModel):
    actions: Union[Episode, State, Segment] = None
    goal: Union[Episode, State, Segment] = None
    optimalilty: Union[float, List[float]] = None

class Description(BaseModel):
    feature_selection: List[gym.Space] = None
    feature_importance: Union[float, List[float]] = None
    feature_ranking: Union[float, List[float]] = None
```

```python
class StandardizedFeedback(BaseModel):
    feedback_id: int = -1
    feedback_timestamp: int = -1
    feedback_type: StandardizedFeedbackType = StandardizedFeedbackType()
    content: Union[Evaluation, Instruction, Description] = None

class AbsoluteFeedback(StandardizedFeedback):
    episode_id: EpisodeID
    target: Union[Episode, State, Segment] = None

class RelativeFeedback(StandardizedFeedback):
    episode_ids: List[EpisodeID] = []
    target: List[Union[Episode, State, Segment]] = []
```