# OpenReview forum: "RLHF-Blender: A Configurable Interactive Interface for Learning from Diverse Human Feedback"
_ICML.cc/2023/Workshop/ILHF — ILHF Workshop ICML 2023_

### Official Review · Reviewer_4zrR · 2023-06-11
**An interesting computational framework for studying RLHF**

**Rating:** 7
**Confidence:** 4

**Review:**

This paper proposes RLHF-blender, an experimental framework for researchers to study human feedback for reward learning. Overall the reviewer thinks this paper is a very good fit for the workshop, as it provides a computational framework for studying how human feedbacks affect RL. The proposed system seems intuitive and easy to follow, which is another plus. As for future improvements, there are also other ways to make this framework more computationally accessible, such as adding an open-source demo or implementation (which the reviewer believes the authors may have already been preparing, from the appendix).

---

### Official Review · Reviewer_66SD · 2023-06-16
**Reads too much like a review without concrete research questions, plan or attempt**

**Rating:** 5
**Confidence:** 4

**Review:**

Summary:
In this paper, the authors proposes that all human feedback types can be characterized by 6 granularities, and outlines a potential interface for blending these feedbacks. The paper reads more like a review than an actual research attempt focusing on a particular RLHF problem, and there is no experiment design or evidence.

Pros:
1. The paper is written in a clear format and generally easy to follow. The color coding of different reward type is also very helpful.

Cons:
1. My major issue is that while the authors identify the existence of multi-type feedback, and positions the paper as an attempt at bridging together these feedback types, there is no proposal to do it backed by theoretical or empirical evidence. The only place that mentions how to do it is in section 6.2 “we might train a multi-objective loss”.
2. The author also proposes in in section 6.3 to perform linear blending on human rationalities (beta) estimated from multiple sources. Again, there is no empirical backings of this.

Suggestions:
While I believe RLHF in practice requires complex programming systems and there is value in system-level research, I would recommend the authors to narrow down on a few specific research questions (i.e. are blending actually helpful? Is alpha blending valid), or specific use cases (i.e. training dm suites using three types of feedbacks, as opposed to others). These questions can be demonstrated and studied on the software platform that the authors develop.

---

### Decision · Program_Chairs · 2023-06-20

Accept